# School Failure in a Girl with Specific Learning Difficulties, Suffering from Childhood Depression: Interdisciplinary Therapeutic Approach

**DOI:** 10.3390/brainsci10120992

**Published:** 2020-12-16

**Authors:** Paraskevi Tatsiopoulou, Georgia-Nektaria Porfyri, Eleni Bonti, Ioannis Diakogiannis

**Affiliations:** 1st Department of Psychiatry, School of Medicine, Faculty of Health Sciences, Aristotle University of Thessaloniki, General Hospital “Papageorgiou”, Ring Road Thessaloniki, N. Efkarpia, 54603 Thessaloniki, Greece; geoporfyri@hotmail.fr (G.-N.P.); elina.bonti@gmail.com (E.B.); idiakogiannis@auth.gr (I.D.)

**Keywords:** major depressive disorder (MDD), specific learning difficulties (SLD), interdisciplinary approach, child psychiatry

## Abstract

Introduction: Recent studies confirm the association of literacy difficulties with higher risk of both childhood behavioral and mental disorders. When co-morbid problems occur, it is likely that each will require separate treatment. The management of major depressive disorder (MDD) for a 9.5 years old girl with specific learning difficulties (SLD), a protracted clinical course, and a family history of affective disorders, was challenging for the interdisciplinary team of our clinic, dealing with learning disabilities. Aim: The research and examination of the first-onset major depressive disorder (MDD) in a child with specific learning disabilities and its impact on school performance. This case report examines the potential contributory factors, but also the recent evidence on the co-morbidity between literacy difficulties and mental illnesses in children. Method: Reporting a two years follow-up of a 9.5 years old child with SLD suffering from childhood depression. Results: A 9.5 years old child with no history of affective disorders, but with a family history of first-degree and second-degree relative suffering from childhood-onset, recurrent, bipolar or psychotic depression. The child was assessed by a child psychiatrist during a period of 2 years, with an average of follow-ups between 1 or 2 weeks. The discussion highlights diagnostic and treatment pitfalls, as well as developmental issues. Practical interventions are suggested. Conclusion: A psychiatrically charged familial environment, including a mother suffering from anxiety disorder and behavioral disorder, contribute significantly to the development of depression in early age. An early medical intervention would be the key for successful treatment. The combination of psychotherapy and antidepressants (mostly selective serotonin reuptake inhibitors (SSRIs)) is the suggested therapy for childhood MDD.

## 1. Introduction

Learning disorder refers to the term Specific Learning Disorder (SLD), which, according to the diagnostic criteria of the Diagnostic and Statistical Manual of Mental Disorder, 5th Edition (DSM-5) [1], is defined as the difficulty in academic skills, such as learning, reading, comprehension and spelling difficulties, written expression difficulties (such as multiple grammar or punctuation errors, inadequate paragraph organization and unclarified written expression), and math difficulties, including calculation and problem solving [1]. The persistence of at least one of these symptoms for more than 6 months, despite specific educational interventions, is diagnostical; while other mental, neurological, intellectual, and physical disorders, such as vision and hearing impairment, should be excluded, as well as social and educational inadequacy. Difficulties in reading and understanding discourage children from learning, intensify school failure, and have negative impact οn self-esteem and mental wellbeing, being associated with varied levels of stress and depression [2].

The childhood major depressive disorder (MDD) is a long-term disease associated with reduced psychosocial functioning [3]. A familial environment charged by psychiatric disorders contributes critically to the development of depression [4]. When onset occurs in childhood, MDD affects both the academic performance and the social interaction which will later be translated to serious difficulties in overall life. It also increases the risk of the manifestation of other psychiatric disorders, such as substance use addictions; additionally premature parenthood is also observed [5,6,7].

The Centers for Disease Control and Prevention estimate that 0.5% of children 3–5 years old, 2% of children 6–11 years old, and up to 12% of children 12–17 years old may test positive for depression [8], without differentiation between boys and girls during childhood [5]. Furthermore, many studies indicate the association of SLD with depressive symptoms [9]. The link between school performance and depression is highlighted by several researchers. School maladjustment was a predictive factor for depression, self-harm, as well as low self-esteem [10,11], while on the other hand, depression was associated with a negative impact on concentration, reading, writing, social relationships, self-reliance in schoolwork, and school performance [12,13].

Although the connection between academic performance and depression has been examined by multiple studies, the research does not describe a causal relationship between the two variables; not clarifying whether depression results to a poor academic performance or, inversely, if problems in academic functioning lead to depression [9]. We would rather picture their relationship as a vicious circle; as depression impairs the child’s self-efficacy and academic performance to the extent that the child cannot believe in its own capacity and tends to perform even poorer academically, aggravating the distress [9]. The manifestation of a significant number of the symptoms of depression usually results in the poor use of learning strategies [9].

Some researchers highlight that children assigned to a special needs education program did not score higher in depression, stress, or maladaptive attributional style than those accustomed to the special-education setting. However, in some studies there were important differences between the combined learning-disability groups and the control group in both stress and depression [14]. Regarding the impact of learning difficulties on children’s psychological well-being and mental development, numerous studies emphasize unnerving results concerning self-harm, suicidality, and depression during childhood, and especially during adolescence [15].

Childhood depression is underdiagnosed and undertreated. Clinical presentation varies but, generally we monitor more somatic symptoms accompanied by a weakness of expression of feelings in preschoolers. Major depressive disorders are monitored in preschoolers from the age of 3 years old. However, depressed preschoolers (3–5 years old), as well as depressed children of middle- childhood (6–8 years old), present mostly physical symptoms, a high irritability, anxiety, signs of depression, and other behavioral problems. In addition, we witness a low self-esteem, a syndrome of guilt, hopelessness, an increased boredom, a need for evasion or escape, and a fear of death. These symptoms are common for the children in school-age, as for those between 9 and 12 years old. Concerning adolescents, the symptoms become more consistent with the DSM-5 criteria [5].

MDD diagnosis, according to DSM-5 [1], is attributed in the presence of at least five symptoms relating to significantly low mood or irritability; moroseness; loss of pleasure or interest in activities that previously enjoyed, nearly every day and during a long part of the day; fatigue; weight loss or weight gain; sleep disorders; psychomotor agitation or retardation; worthlessness; exorbitant to delusional guilt; inattention; indecision, almost every day; unwanted, intrusive thoughts about death or fear of death. The symptoms cause impairment in school attendance and social functioning in children suffering from at least one of the symptoms of depressed mood or loss of pleasure or interest [1].

Early intervention is recognized as a key element for the treatment of depression in children and adolescents. Psychotherapy is usually the initial treatment (for a mild to moderate depression), followed by antidepressant medication (for moderate to severe depression) [5].

Children who have received treatment might present a relapse within 2 years in a percentage of 70% [5,16,17]. However, no single variable has been proved as a predictor of recurrence of MDD in children. Factors that may predict relapse include early onset; a high number of depressive episodes, as well as their severity; and finally, psychosocial stressors combined with a co-morbid persistent depressive disorder [5,18,19]. Additionally, a familial environment charged by psychiatric disorders may significantly contribute to the risk of developing and recurrence of depression.

## 2. Materials and Methods

Through a 2 years follow-up, we proceed to the report of a clinical case of a 9.5 year old girl with specific educational disabilities (SLD), who also experienced a major depressive disorder (MDD).

## 3. Results

The girl was first diagnosed with specific learning disabilities (SLD) at the age of 8. Her parents, at the instigation of the schoolteacher, decided to proceed at a learning assessment when she was in the middle of the second grade. Initially she was addressed to a special educator, who conducting a standardized learning assessment, test Athena [20], diagnosed SLD. The Athena test, has been standardized for children between 5 to 9 years old and evaluates oral expression, listening comprehension, written expression, reading skills, reading comprehension, spelling, mathematics calculations, and mathematics reasoning [20].

After almost a year and a half, the 9.5 year old girl was subjected to a pediatric consultation, as her parents noticed that she was no longer playing with toys or with friends. The Child and Adolescent Psychiatric Assessment (CAPA), which usually takes place in our medical setting, in collaboration between the Outpatient State Certified Diagnostic Department for Learning Difficulties and the Child Psychiatry, highlighted a range of psychological symptoms that were common in depression, while the learning assessment confirmed the SLD.

The learning assessment included both academic and cognitive assessment, as well as the recording of developmental history. Regarding the academic assessment, an atypical (non-standardized) learning assessment was conducted, which is due to be standardized, consisting of a number of tasks evaluating basic non-curriculum based academic skills in the areas of literacy, language, and mathematics that provide a full, sufficient and clear picture of the different specific academic skills of the individual evaluated, albeit within a short period of time [21]. The atypical learning assessment, in comparison with the Athena test, also indicated SLD. The results showed that the girl had difficulties mainly in reading, spelling, and written expression. More specifically, she was reading well below the expected level for her age and displayed omissions, substitutions, distortions, or additions while reading. She had difficulty seeing and hearing similarities and differences in letters and words, and she was unable to sound out the pronunciation of an unfamiliar word. She was also avoiding activities that involved reading and writing, she was spending an unusually long time completing tasks that involved reading or writing, and she had difficulty finding the right word to express what she wanted and structuring her sentences.

For the cognitive assessment, the Wechsler Intelligence Scale for Children Forth Edition (WISC-IV) (for ages 6 to 16) was used [22]. Intellectual functioning in WISC-IV showed a significant discrepancy between the General Ability Index (GAI) (score = 107) and the Cognitive Proficiency Index (CPI) (score = 92). The score was higher in the Verbal Comprehension index (VCI) and Perceptual Reasoning index (PRI), and lower in the Working Memory index (WMI) and Processing Speed index (PSI). Specifically, coding subtests, similarities, digit span, letter-number sequences recorded worse performances. The girl’s full-scale Intelligence Quotient (FSIQ) was average, between 90–108. In order to be diagnosed with SLD, the IQ of the child must be average or higher.

The developmental history of the girl showed that, even though the more serious learning difficulties were perceived at the beginning of formal schooling, first grade, some signs already existed from preschool, mainly an attention impairment that continued in her school years. At preschool, the girl also had problems forming words correctly, such as reversing sounds in words or confusing words that sounded alike, as well as remembering or naming letters, numbers, and colors. She was learning new words slowly and she had difficulties in learning nursery rhymes or playing rhyming games. She was not diagnosed and did not show signs of any other developmental disorder (i.e., speech delay; ADHD). Her parents did not mention any abnormality in her milestones. According to them, she started saying words, such as ‘mama’ and ‘dada’, at 12 months old and she started using more complete phrases at 30 months old. Her mother mentioned that her daughter was late at developing her vocabulary when compared to her older brother at the same age, but did not feel concerned at the time. Regarding the motor developmental milestones, the girl started to crawl when she was 9 months old and started to walk alone when she was 15 months old.

Regarding CAPA, parents mentioned that, for about 1.5 month before the consultation, the child was complaining of several headaches and stomach aches, as well as of fear that her parents were going to die. Also, there had been irregular periods of uncontrollable outbursts of crying and shouting with no known cause. In parallel, for several weeks, school teachers had noticed inattention in the classroom and an increasing isolation and irritability without any obvious reason. Both the parents and teachers were asked to complete a series of questionnaires, including the Hellenic ADHD Rating Scale-IV, for parents and teachers [23], the Strengths and Difficulties Questionnaire (SDQ) -Hel-Emotional Symptoms Subscales [24] and the DSM-5 diagnostic criteria for ADHD Inattention Subscale, which contains nine items used to define the inattention symptomatology of the ADHD diagnosis, according to DSM-5 [1,23,24,25]. According to the data from previous studies, a score above 95th percentile is indicative of the presence of mood disorders [26]. The SDQ impact score evaluated the impact of inattention and emotional problems on everyday functioning [27]. As both teachers and parents evaluated that the girl had emotional and social problems, they gave additional information through reports concerning different settings, such as family life, interpersonal relationships, friendships, learning in school, and entertainment activities both at school and home. Our patient had a high score in Inattention and Emotional Symptoms Subscales, indicating the impairment in her daily functionality.

Despite all, the family history revealed a high-risk. One first-degree relative (oldest brother) and one second-degree relative (grandmother), had registered a childhood-onset, recurrent bipolar or psychotic depression, while the mother of the patient suffered from an anxiety disorder. This high risk family history of psychiatric and behavioral disorders significantly raised the risk of developing reoccurring depression.

The girl was followed up over the course of 2 years, with an average interval between the consultations of 1 to 2 weeks. The 2 years follow-up gave us the opportunity to research and apply various options of treatment concerning childhood depression, such as psychoeducation, family education, and psychotherapy (individual and psychodynamic) combined with medication. Psychotherapy included an individual psychodynamic psychotherapy, in a frequency of 1 or 2 sessions per week for 2 years [28]. Daily administration of 10 mg of fluoxetine over 10 weeks, later increased to 40mg over 4 weeks, was not effective regarding the symptoms of depressed mood, dysthymia, insomnia, and school failure. On the contrary, therapy was effective regarding the capability of establishing friendships, and also regarding feelings that life as well as school attendance have an importance. The patient denied any attempt of self-injury. Adverse effects from fluoxetine did not occur and thus there was no need in trying another antidepressant medication.

## 4. Discussion

At the beginning of the psychiatric intervention, the mild to moderate depressive episodes were treated with a multidisciplinary approach, including psychotherapy, psychoeducation, and family education. After the two year follow-up and the therapy intervention a more severe depressive episode required pharmacological treatment.

Psychoeducation was crucial for each part, so that both patient and parents would be aware of the treatment context and the first goals. Regarding the awareness of the situation, the psychoeducation that was provided to the parents included the warning signs and symptoms of depression, the illness trajectory, the risk of relapse, the therapy choices, and advice about interacting with their depressed child [18].

Although, pharmacological treatment with antidepressants may be considered as the initial therapy for a patient not responding to psychotherapy, for moderate to severe depression [5,29,30,31], it should not be the only form of therapy for childhood depression, but should be combined with psychotherapy [5]. Selective Serotonin Reuptake Inhibitors are used as the initial antidepressant treatment during childhood and adolescence MDD. Fluoxetine has shown strong evidence of effectiveness in childhood MDD and is approved for middle-childhood depression, from the age of 8 years old, by the Food and Drug Administration (FDA) [5,32,33,34,35].

Suicidal behaviors are common in children with MDD [5]. Risk factors of higher suicidality in children and adolescents include previous self-harm attempts, as well as poor parental involvement in therapy [5]. The success of pharmacological administration in childhood MDD is evaluated by the occurrence of low suicidality risk, the parental supportive involvement, the ability of parents to observe the signs and symptoms and to cope with the difficulties of their child regarding the treatment; and finally, the capacity of mental health professionals to monitor the patient’s therapeutic trajectory [28,32].

In conclusion, psychotherapeutic sessions combined with pharmacological treatment is the most commonly suggested approach for the treatment of childhood depression that is characterized by a high risk of suicidality, or of a switch to hypomania or mania [5]. The appropriate medical monitoring includes a close observation of any change in behavior, and of any suicidal thoughts or actions. Although the FDA no longer suggests a specific follow-up schedule, a follow-up evaluation in 4 to 6 weeks, if not sooner, is suggested in clinical practice [5,32].

## 5. Conclusions

Inattention in childhood, expressed through lack of concentration and incongruous behaviors that do not correspond to parents’ and teachers’ expectations, as well as school requirements [36], may have a negative impact on child’s functionality both at school and family context [23,37,38]. According to DSM-5 diagnostic criteria, SLD and various mental disorder manifestation in childhood may concern symptoms such as difficulties in concentration [1,23,36,39]. The simultaneous consideration of information issued both from parents and teachers resulted in the depression symptomatology including physical symptoms, irascibility with no known reason, emotional vulnerability, yelling, loss of pleasure, and seclusion. Without the parallel examination and verification of information from the two sub-mentioned sources, such symptoms could had been considered as resulting from ADHD or mood, behavioral and other disorders, or various other conditions, such as bullying, given that there was no mention of any stressful event or life changing event, recent loss, trauma or treatment with psychotropic medication [24].

Even if many studies revealed that depression affects a major percent of youth with learning problems, highlighting the association between childhood MDD and academic performance, the variability and difference of symptoms as they are presented to adult cases makes it under-recognized and undertreated. Consequently, poor academic performance, social dysfunction, and an increased suicide risk could potentially pass without the necessary attention. The high familial charge concerning psychiatric disorders significantly raise the risk of the development and recurrence of depression, and must be taken under consideration. Diagnosing childhood depression is quite complex. Thus, the observations of parents and teachers are of great importance, as they keep close supervision of the child’s life and daily activities, and the depression has a negative effect on interpersonal, school and home life. The first line treatment of mild symptomatology of childhood depression includes psychotherapy, psychoeducation, and family involvement, while a combination of psychotherapy and antidepressant pharmacological treatment is recommended to confront moderate to severe childhood depression.

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
