# Peer review of "School Failure in a Girl with Specific Learning Difficulties, Suffering from Childhood Depression: Interdisciplinary Therapeutic Approach"

_brainsci, 2020, doi:10.3390/brainsci10120992_

Round 1
Reviewer 1 Report
The manuscript is entitled " School failure in a girl with specific learning difficulties, suffering from childhood depression: interdisciplinary therapeutic approach. Case Report. ". The objective is to investigate the potential contributory factors on the co-morbidity between literacy and difficulties and psychiatric disorders such as depression in a child using a longitudinal method.
The manuscript is clear and well-conducted.
Just a comment : Line 61, it could be pertinent to propose some references to illustrate the link between depression and school performance.
Author Response
Response to Reviewer 1:
Comment: Line 61, it could be pertinent to propose some references to illustrate the link between depression and school performance.
Thank you for your comment, it was extremely useful, as the link between school performance and depression is highlighted by several researchers. We have incorporated information in lines 60-65: “Furthermore, many studies indicate … and school performance [12,13]”.
Reviewer 2 Report
In the manuscript, the authors reported an interestingly description on the diagnostic workout and follow-up of a pediatric case, presenting with mood disorder and comorbid learning difficulties, giving interesting point of view on the therapeutic approach, with a combined treatment with medical follow-up, psychodynamic therapy and pharmacotherapy
In the Material and Methods- Case report 2.2., the psychiatric evaluation is reported by several diagnostic tolls, such as SNAP-IV and SDQ while the DSMV criteria for the ADHD have been highlighted for inattention problems. But there is not mention of further evaluations for Specific Learning Disabilities or about the Cognitive profile (WISC-IV). I’d like to suggest expanding the description about the over mentioned data and about the school course of the child (i.e. when learning difficulties onset and if they were specific or isolated to an attention impairment). Moreover clinical data on the Neurodevelopment of this child (i.e. speech; motor developmental millestones) could be important in order to evaluate a Neuro-typical profile or developmental disorders (Speech delay; ADHD)
In conclusion, I would recommend these major revisions to the manuscript.
Moreover in the manuscript I would suggest further revisions as following:
- Line 21 I’d suggest to change “genetic history” with “family history”
- Line 124-128: I’d like to suggest to report this comment in the discussion but not in the case report description, as the line from 152 to 158 ( till".. changes in daily life)
- Line 161-162 (The girl was followed….. of 1 to 2 weeks) postpone this phrase before the line 165 (..The two years follow-up), getting focused on the therapeutic course of this case.
- Line 182-183: This description could be included in the methods, describing the therapeutic approach used, at the line 165.
- English revision: line 67 “The more significant the number of symptoms of depression, the lower the use of learning strategies”; Line 103 “The less or more earlier age of onset”; Line 168 “didn’t annulated”; Line 179 “by the two parties”
Author Response
Response to Reviewer 2:
A-Major Comments: In the Material and Methods- Case report 2.2., the psychiatric evaluation is reported by several diagnostic tolls, such as SNAP-IV and SDQ while the DSMV criteria for the ADHD have been highlighted for inattention problems. But there is not mention of further evaluations for Specific Learning Disabilities or about the Cognitive profile (WISC-IV). I’d like to suggest expanding the description about the over mentioned data and about the school course of the child (i.e. when learning difficulties onset and if they were specific or isolated to an attention impairment). Moreover clinical data on the Neurodevelopment of this child (i.e. speech; motor developmental millestones) could be important in order to evaluate a Neuro-typical profile or developmental disorders (Speech delay; ADHD)
We agree that this is a very important part and we thank you for pointing it out. We expanded the results (3), by providing information on:
- a) the evaluation for Specific Learning Disabilities: lines 114-118: “a standardized Learning … mathematics reasoning [20]”; and lines 126- 140: “Learning assessment included … to structure her sentences”.
- b) the Cognitive profile (WISC-IV): lines 141- 149: “For the cognitive assessment… be average or higher.”
- c) the school course of the child: lines 150- 157: “The developmental history of… disorder (i.e. speech delay; ADHD).”
- d) the neurodevelopment of the child, concerning speech and motor developmental milestones: “Her parents … when she was 15 months old”: Lines 157-163: “Her parents did not mention … she was 15 months old.”
B- Minor comments:
We have restructured the following sentences according to your suggestions:
- Line 21 I’d suggest to change “genetic history” with “family history”: Line 25
- Line 124-128: I’d like to suggest to report this comment in the discussion but not in the case report description, as the line from 152 to 158 ( till".. changes in daily life): We reported this comments in the conclusion: Lines 228-239.
- Line 161-162 (The girl was followed….. of 1 to 2 weeks) postpone this phrase before the line 165 (..The two years follow-up), getting focused on the therapeutic course of this case:
We have entirely restructured this paragraph according to your suggestion: lines 186- 191: “The girl was followed up … for the last 2 years [28].”
- Line 182-183: This description could be included in the methods, describing the therapeutic approach used, at the line 165.
We included this description in the results, describing the therapeutic approach used, at the line 189-191 “Psychotherapy included… for the last 2 years [28].”
- English revision:
- Line 67 “The more significant the number of symptoms of depression, the lower the use of learning strategies”: Revised as “The manifestation of significant number of symptoms of depression, usually results to poor use of learning strategies”; Lines: 72-73.
- Line 103 “The less or more earlier age of onset”: Revised as “early onset”; Line 102.
- Line 168 “didn’t annulated”: Revised as “wasn’t effective regarding”; Line 192.
- Line 179 “by the two parties”: Revised as “both patient and parents”; Line 202.
Round 2
Reviewer 2 Report
I agreed with the changes made in the text and I have no further comments to add